# VIRL: Self-Supervised Visual Graph Inverse Reinforcement Learning

**Lei Huang**[1], **Weijia Cai**[1], **Zihan Zhu**[2], **Chen Feng**[3], **Helge Rhodin**[4,5], **Zhengbo Zou**[1]
[1]Columbia University [2]Brown University [3]NYU [4]UBC [5]Bielefeld University

**Abstract:** Learning dense reward functions from unlabeled videos for reinforcement learning exhibits scalability due to the vast diversity and quantity of video resources. Recent works use visual features or graph abstractions in videos to measure task progress as rewards, which either deteriorate in unseen domains or capture spatial information while overlooking visual details. We propose **V**isual-Graph **I**nverse **R**einforcement **L**earning (VIRL), a self-supervised method that synergizes low-level visual features and high-level graph abstractions from frames to graph representations for reward learning. VIRL utilizes a visual encoder that extracts object-wise features for graph nodes and a graph encoder that derives properties from graphs constructed from detected objects in each frame. The encoded representations are enforced to align videos temporally and reconstruct in-scene objects. The pretrained visual graph encoder is then utilized to construct a dense reward function for policy learning by measuring latent distances between current frames and the goal frame. Our empirical evaluation on the X-MAGICAL and Robot Visual Pusher benchmark demonstrates that VIRL effectively handles tasks necessitating both granular visual attention and broader global feature consideration, and exhibits robust generalization to *extrapolation* tasks and domains not seen in demonstrations. Our policy for the robotic task also achieves the highest success rate in real-world robot experiments. Project website: https://leihhhuang.github.io/VIRL/.

**Keywords:** Inverse Reinforcement Learning, Learning from Video

## 1 Introduction

Intelligent agents have mastered complex tasks by learning policies through reinforcement learning (RL), particularly in the gaming domain, where their performance can match or even surpass champion level [1, 2, 3]. Constructing a precise dense reward function is fundamental and indispensable for appropriately training an agent for a task, as it succinctly defines the task to master [4] and provides dense signals for faster policy learning [5, 6].

However, hand-designing such functions requires substantial domain knowledge and extensive engineering efforts, making it unscalable given the myriad tasks in the real world [7]. Moreover, the process of fine-tuning reward functions [8] is susceptible to overlooking corner cases, leading to agent misbehavior due to unintended sub-optimal rewards [9]. Recently, large language models (LLMs) have been leveraged to assist in generating executable reward functions for RL [10, 11], but they necessitate appropriate prompts and function templates from domain experts and involve considerable training time due to the evolutionary search process.

On the other hand, readily available video demonstrations contain implicit dense rewards in the form of continuous progress towards goals. After watching videos, one can easily estimate how much progress is achieved given a frame and the goal. To this end, inverse reinforcement learning (IRL) has been proposed [12, 13, 14, 15, 16] to learn reward functions from demonstrations by utilizing features of environments and agents. Compared to those requiring ground-truth actions [17, 18] or

8th Conference on Robot Learning (CoRL 2024), Munich, Germany.

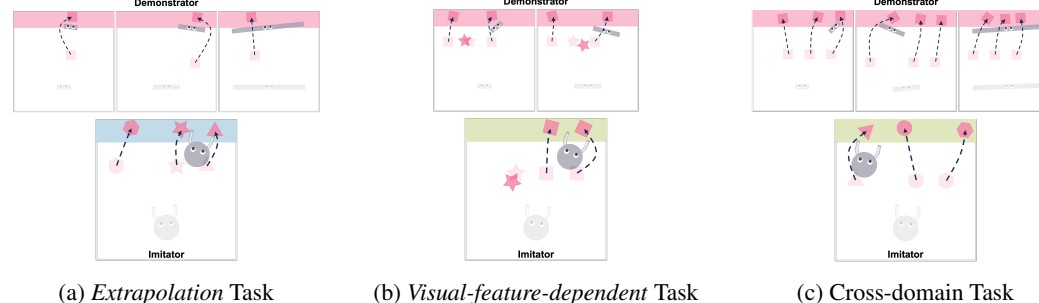

|  (a) *Extrapolation* Task | (b) *Visual-feature-dependent* Task | (c) Cross-domain Task |

Figure 1: Capabilities of VIRL: (a) Generalizable to task **extrapolations**; (b) Adaptable to **visual-feature-dependent** task, such as pushing debris with identical shapes; (c) Robust to **domain shift**. Figures on the top are unlabeled video demonstrations and those on the bottom are tasks for imitators to learn. X-MAGICAL benchmark contains 4 embodiments: short-stick, medium-stick, long-stick as shown in (a) top from left to right, and gripper in the bottom. All tasks are set to be cross-embodiment. **Note**: Long-stick agent is *unsuitable* for (b) visual-feature-dependent tasks due to its excessive length, which hinders task execution.

inferred actions from dynamic models [19, 20], IRL algorithms for action-free videos [21, 22, 23] have garnered greater attention recently due to their scalability and lower cost of data collection.

Recent IRL works [23, 24] achieve impressive performance in building task-specific embodiment-agnostic reward functions using encoders pretrained by temporally aligning videos of varied lengths [25]. Agents are encouraged to reach the provided goal in the latent space by receiving rewards based on the distance between the embeddings of the current state and the goal state. However, employing visual encoder [23] suffers from performance deterioration in unseen domains; and utilizing graph abstractions of object coordinates [24], while robust to domain shifts, sacrifice visual features, which can be crucial for certain tasks. Additionally, since their representations are task-specific, prior works have not been tested beyond demonstrated tasks, such as extrapolations, which may pose challenges even with a minor change in object counts [26]. Therefore, we are interested in two questions: (i) How can we leverage both low-level visual features and high-level graph abstractions to adapt to both geometry-aware and -unaware tasks? (ii) Can a single pretrained encoder be used to build reward functions of tasks beyond those it is trained on, such as task extrapolations?

To this end, we propose a visual graph encoder that combines a visual encoder for capturing object-centric low-level features with a graph encoder for extracting scene-level abstractions. By formulating object-centric graphs that are generalizable across various object counts, we demonstrate that the pretrained visual graph encoder can construct reward functions for tasks of manipulating an unseen number of objects. This capability is inspired by the adaptability and flexibility of graph neural networks to encode graph structures with varying numbers of nodes. Figure 1 illustrates the main properties of our approach, and Table 2 in Appendix A.1 compares advantages over related works.

## 2 Related Work

For a more comprehensive related work, please see Appendix A.2.

**Imitation learning with actions.** Imitation learning from expert demonstrations with state and action pairs has demonstrated successes in various approaches [27], from behavioral cloning [28, 29, 30] to inverse reinforcement learning [17, 31, 32]. However, its reliance on ground-truth actions renders data acquisition at scale both costly and challenging, thereby limiting its wider applicability. To leverage unlabeled video demonstrations that are easier to obtain, methods have been developed to infer actions in videos [33, 34, 35, 36, 37, 38, 39]. For example, VPT [39] trains an inverse dynamics model (IDM) on a small labeled dataset and uses the trained IDM to augment a large number of unlabeled videos with actions, which are used for the subsequent policy learning via behavioral cloning. These methods have successfully trained agents using unlabeled videos, but are

subject to well-defined morphologies and dynamics of demonstrators and imitators, making it hard to leverage demonstrations with diverse embodiments.

**Domain adaption for imitation learning.** To overcome the domain gap between varied agent embodiments, several works [40, 41, 42, 43, 44, 18] attempt to translate the context in demonstrations to the imitators' context in terms of agents and viewpoints, using methods such as generative adversarial networks [45, 46] and context translation. Furthermore, another way to address the domain gap between agents is by inpainting them from video demonstrations and online visual states [47].

**Inverse reinforcement learning from videos.** On the other hand, a group of works [21, 22, 48, 49, 50, 44, 51, 52, 53, 24] has focused on learning reward functions from videos by extracting latent features that indicate intermediate steps or measure task progress, which are naturally and implicitly present in demonstrations. Learned reward functions are then used for policy training in the regular RL regime. To train on videos with varied lengths and paces, XIRL [23] adopts temporal cycle-consistency loss [25] to learn visual embeddings that measure task progression and then uses distances between current state embedding and the goal embedding as reward signal for policy learning. They demonstrate the pretrained encoder benefits from the diversity of demonstrators' embodiments and generalizes to unseen embodiments. Building on XIRL, our work seeks to train an encoder that generalizes to constructing reward functions for task extrapolations by leveraging graph abstractions, while maintaining the flexibility for tasks requiring attention to low-level visual features by encoding object features.

**Object-centric scene graphs.** Abstracting visuals into object-centric scene graphs is beneficial for agents to understand environments and make decisions in various tasks [54], such as dynamics modeling [55, 56], navigation [57, 58, 59], and robotic manipulation [60, 61, 62, 43, 63, 64]. Following up XIRL, GraphIRL [24] enhances robustness to diverse object appearances by abstracting frames into graphs, where each node represents an object and contains features including its bounding box coordinate and distances to other objects. However, limitations arise when tasks require attention to object geometries or when dealing with different numbers of objects. Motivated by their work and aiming to address the limitations, we encode object visual features and locations into nodes in a unified way and build graphs such that the encoder can generalize to scenes with arbitrary numbers of objects. Our work can be considered a generalized version of GraphIRL, suitable for both geometry-aware and -unaware tasks, as well as unseen task extrapolations.

## 3   Method

Our objective is to learn reward functions from video demonstrations showcasing a specific task, enabling an agent with a different embodiment to learn the task and its extrapolations where there is an unseen number of objects to manipulate (Section 3.1). Our approach involves two phases: Firstly, pretraining a visual graph encoder (Section 3.2) to capture object-wise visual features and global scene features in a self-supervised manner (Section 3.3); Secondly, cross-embodiment policy learning for tasks by training RL algorithms with dense reward functions built on the pretrained encoder (Section 3.4). The overall framework of VIRL is depicted in Figure 2.

### 3.1   Problem formulation

VIRL takes as input a dataset of action-free videos $D^T = \bigcup_{k=1}^{K} V_k^{e,T}$ of the identical task $T$, where $V_k^{e,T}$ is the $k^{th}$ video demonstration conducted by an agent with embodiment $e$. A video is defined to be a sequence of image frames $V_k^{e,T} = \{I_1^k, I_2^k, ..., I_{L_k}^k\}$, where $L_k$ is $k^{th}$ video's length. We design a visual graph encoder $\phi_{vg}$ to transform $D^T$ into a dataset of graph sequences $G^T = \bigcup_{k=1}^{K} G_k^{e,T}$, where $G_k^{e,T} = \{\mathcal{G}_1^k, \mathcal{G}_2^k, ..., \mathcal{G}_{L_k}^k\}$, and extract graph encodings. We pretrain the encoder via temporal alignment of graph encoding sequences and pixel reconstruction of manipulable objects. The pretrained encoder is then utilized to construct general-form dense reward functions that gauge task progress through low-level visual and high-level structural features in graphs. It merits attention that the visual graph encoder's versatility allows it to not only formulate a reward function for the

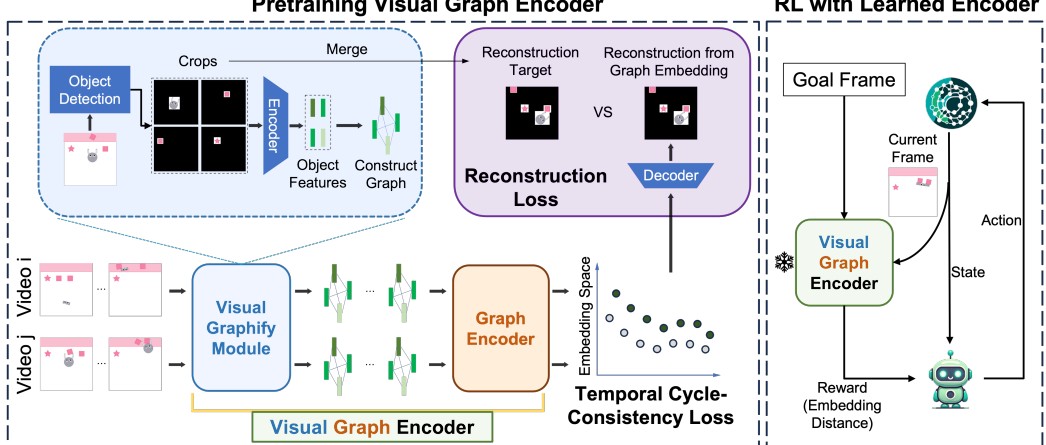

Figure 2: Overview of Visual Graph Inverse Reinforcement Learning. We transform videos to graph sequences containing both low-level visual features and high-level global features, pretrain the visual graph encoder via temporal alignment and object-centric scene reconstruction, and build reward functions using the visual graph encoder for unseen embodiment to learn the demonstrated task and task extrapolations with reinforcement learning.

demonstrated task $T$ but also for its *extrapolation* tasks $T'$ in a zero-shot fashion, attributable to the encoder's capability to process variable-sized graphs. We define extrapolation tasks $T'$ to be those where agents manipulate more objects than demonstrated, which are unseen in videos.

## 3.2 Visual graph encoder

We propose a visual graph encoder composed of a visual graphify module and a graph encoder. The visual graphify module builds object-centric graphs from images by encoding the spatial and visual features of each object in an image into graph node features and objects interactions into edge features. Then, the graph encoder takes in graphs and outputs the representations, which will be trained via temporal alignment in the latent space and reconstruction of object-centric frames.

**Visual graphify module.** We use the proposed visual graphify module to transform every sequence of frames $V_k^{e,T} = \{I_1^k, I_2^k, ..., I_{L_k}^k\}$ in the dataset $D^T$ to a sequence of graphs $G_k^{e,T} = \{\mathcal{G}_1^k, \mathcal{G}_2^k, ..., \mathcal{G}_{L_k}^k\}$ to get $G^T$. A graph is defined as $\mathcal{G}_t^k = (\mathcal{V}, \mathcal{E})$, where $\mathcal{V}$ is the node set corresponding to objects in the frame and $\mathcal{E}$ is the edge set indicating the relationships between objects. For every frame $I_t^k \in \mathbb{R}^{H \times W \times C}$, an off-the-shelf detector is employed to get $N$ bounding boxes of $N$ objects in the scene. Contrary to approaches that extract visual and positional features using a CNN-based encoder and an MLP separately [65, 66], we encode both features from regions defined by bounding boxes using a shared CNN-based encoder in an unified way. Specifically, for each object in an image, the pixels within its bounding box is kept while all the other pixels being masked so that the *positional* information is encoded implicitly. After this operation, an image is transformed to a stack of images $I_{t,o}^k \in \mathbb{R}^{N \times H \times W \times C}$, each of which contains only one object and is then encoded into an object feature $\mathbf{f}_{o_i} \in \mathbb{R}^{D_{obj}}$ as the feature of a node in $\mathcal{V}$ using the shared ResNet-based [67] visual encoder. The distance between two objects $d_{o_{ij}} \in \mathbb{R}$ represents the edge feature connecting the node pair. We consider all graphs to be fully connected and undirected, allowing for direct interactions between all objects.

Apart from the transformation from images to graphs, we obtain the reconstruction target from the stack of object images $I_{t,o}^k$ to ensure that the embedding after the graph encoder can recover sufficient low-level visual features of objects. It contributes to our work's advantage of not only imitating the spatial interactions between objects but also capturing the low-level visual features of manipulative objects. For example, in the context of the Sweep-to-Goal task in X-MAGICAL [23], a variety of

task variants can be generated, such as pushing multiple objects irrespective of their appearance and selectively pushing duplicates out of three objects present in a scene. With this motivation, we merge all non-agent images in a stack, i.e., $I_{t,o}^{k}{}' \in \mathbb{R}^{(N-1) \times H \times W \times C}$ into a single RGB image to generate the reconstruction target $I_t^{k'}$ for the original frame $I_t^k$.

**Graph transformer encoder.** Subsequent to obtaining $G^T$, we use a graph network to extract frame embeddings for temporal alignment and reconstruction. Considering the impressive performance of Transformer models in computer vision and natural language processing [68, 69, 70], we employ a Graph Transformer [71] network as the encoder. Given a graph with node features $H = \{\mathbf{h}_1^l, \mathbf{h}_2^l, ..., \mathbf{h}_N^l\}$ and edge features $E = \{\mathbf{d}_{ij} | i, j \in [N] \wedge i \neq j\}$, we update the node features using the graph transformer operator by:

$$\mathbf{h}_i^{l+1} = \mathbf{W}_1^l \mathbf{h}_i^l + \sum_{j \in \mathcal{N}_i} \alpha_{ij}(\mathbf{W}_2^l \mathbf{h}_j^l + \mathbf{W}_3^l \mathbf{d}_{ij}), \text{where } \alpha_{ij} = \sigma \left( \frac{(\mathbf{W}_4^l \mathbf{h}_i^l)^\top (\mathbf{W}_5^l \mathbf{h}_j^l + \mathbf{W}_3^l \mathbf{d}_{ij})}{\sqrt{d}} \right).$$

All $\mathbf{W}$s are different trainable parameters. $\alpha_{ij}$ is the attention coefficient and $\sigma$ is the softmax function. After the graph transformer operations, we apply a global mean pooling layer to extract the average of node encodings as the final embeddings for temporal alignment and reconstruction.

### 3.3 Loss function for pretraining

The visual graph encoder is trained in a self-supervised way. Given each video $V$, we obtain a sequences of graph embeddings $\phi_{vg}(V)$ using the proposed visual graph encoder. For simplicity, we omit subscripts and superscripts for video index $k$, embodiment $e$ and task $T$. We adopt TCC loss [25] to temporally align the graph embeddings between sampled videos for measuring task progress. Meanwhile, we enforce the embeddings to keep objects' low-level visual features by adopting a reconstruction loss, which is inspired by He et al. [72] and Ye et al. [62].

**Temporal alignment.** Given a pair of randomly sampled videos $U$ and $V$, the visual graph encoder computes their embeddings $\phi_{vg}(U) = (u_1, u_2, ..., u_{L_U})$ and $\phi_{vg}(V) = (v_1, v_2, ..., v_{L_V})$. To compute the TCC regression loss, we randomly select an embedding $u_i$ in $\phi_{vg}(U)$ and find its soft nearest neighbor $\widetilde{v}$ in $\phi_{vg}(V)$ by estimating the similarity between $u_i$ and every embedding in $\phi_{vg}(V)$ and conducting a weighted average over $\phi_{vg}(V)$:

$$\widetilde{v} = \sum_j^{L_V} \alpha_j v_j, \text{ where } \alpha_j = \frac{e^{-||u_i - v_j||^2}}{\sum_k^{L_V} e^{-||u_i - v_k||^2}}.$$

We then operate cycle back from $\widetilde{v}$ and find the index of $\widetilde{v}$'s soft nearest neighbor in $\phi_{vg}(U)$:

$$\widetilde{\mu} = \sum_k^{L_U} \beta_k k, \text{ where } \beta_k = \frac{e^{-||\widetilde{v} - u_k||^2}}{\sum_j^{L_U} e^{-||\widetilde{v} - u_j||^2}}.$$

Since the index of the starting point $u_i$ is known to be $i$, the loss can be computed using the squared distance $(\widetilde{\mu} - i)^2$. Additionally, variance $\sigma^2 = \sum_k^{L_U} \beta_k (k - \widetilde{\mu})^2$ is added to the loss function as a regularization term to make predictions less dispersed [25]. Thus, the final objective function for temporal alignment is $L_{tcc} = \frac{(i - \widetilde{\mu})^2}{\sigma^2} + \lambda \log(\sigma)$, where $\lambda$ is a constant weight.

**Object-centric reconstruction.** When coming to tasks that require reasoning about low-level features, such as pushing duplicates among several objects, we want the visual graph encoder to keep as much object-centric low-level information as possible [73]. Thus, we define a decoder $\text{Dec}(\phi_{vg}(V))$ that maps from graph embeddings back to frames. We would like to highlight that, instead of reconstructing the original frame $I_t^k$, our reconstruction target is $I_t^{k'}$ generated in the *visual graphify module* as introduced in Section 3.2, which keeps pixels within the bounding boxes of $N-1$ objects, excluding the agent. We define the reconstruction loss as $L_{rec} = ||I_t^{k'} - \text{Dec}(\phi_{vg}(I_t^k))||_2$.

Finally, we combine the two losses to get the overall loss function for pretraining the visual graph encoder: $L = L_{tcc} + \lambda_1 L_{rec}$, where $\lambda_1$ is a weight term.

### 3.4 Reinforcement learning with learned reward

We use the visual graph encoder $\phi_{vg}$ pretrained on videos of task $T$ to build reward functions for $T$ and its extrapolations $T'$, which are then used for an agent with an embodiment different from demonstrators to learn the tasks with RL algorithms. Since the embedding sequences are temporally aligned in the embedding space, we can build a reward function for a task by measuring the current progress in the embedding space [23], i.e., the negative distance between the current frame embedding $\phi_{vg}(I)$ and the goal embedding $g$: $r(I) = -1/s||\phi_{vg}(I) - g||_2^2$, where $g$ is defined by the average embedding of all last frames in videos, and $s$ is a parameter that rescales the reward to a reasonable range for policy learning [74]. For the most challenging extrapolation task where agents learn to push 3 debris from videos of pushing 1 debris, combining our approach with a reformated reward function: $r(I) = -1/s \times \log(||\phi_{vg}(I) - g||_2^2 + 1)$, can achieve better performance.

Compared to the most related prior works [23, 24], the main advantages of building reward functions using our approach are twofold: (i) it reserves the ability to learn tasks that require attention to low-level features while maintaining generalization to unseen domains by leveraging graph abstraction, (ii) apart from the demonstrated task $T$, it generalizes better to task extrapolations $T'$ because of the flexibility of the encoder to consume graphs consisting of various numbers of objects.

## 4 Experiments

We conduct experiments on the *Sweep-to-Goal* task in X-MAGICAL [75, 23] and Robot Visual Pusher [44, 24] benchmark to answer following questions: (i) Can a pretrained visual graph encoder construct reward functions for extrapolations of the demonstrated task in zero-shot? (ii) Does VIRL retain low-level visual features for learning? (iii) How does VIRL perform in shifted domains [1]? We also deploy policies trained in simulation on a real robot for robotic tasks. For implementation details, please see Appendix A.3.

### 4.1 Experiment setup

**Baselines.** We compare our work against self-supervised approaches that learn reward functions from unlabeled videos: (i) XIRL [23] pretrains a task-specific visual encoder using TCC loss on videos and builds reward functions using the pretrained encoder. (ii) GraphIRL [24] transforms videos to graphs of object coordinates, uses a spatial interaction graph encoder to extract features, and follows a similar pipeline of XIRL. (iii) TCN [22] uses triplet loss to pretrain representations on videos such that frames at the same time are close in the embedding space while those that are temporally different are far apart. (iv) LIFS [73] trains an invariant feature space to transfer skills between agents with different embodiments using a contrastive loss and a reconstruction loss.

**Task descriptions.** *Sweep-to-Goal* in X-MAGICAL. All tasks are set to be cross-embodiment, i.e., an agent learns from videos of other 3 agents. Task I (Figure 1a): learning to sweep 3 debris to the colored goal region on the top, given videos of sweeping 1 or 2 debris. Task II (Figure 1b): learning to sweep 2 debris with the same shape to the goal region while leaving the rest outside, given videos of the same task. Task III (Figure 1c): learning to sweep 3 debris regardless of shapes and colors, given videos of sweeping 3 squares. *Robot Visual Pusher* tasks. Task I (Figure 5): xArm learning to push an object to the goal, given videos of the same task by human. Task II: xArm learning to push 1 specific object out of 2 objects in-scene to the goal, only given the set of human videos in Task I.

### 4.2 Experiment results on X-MAGICAL

**Results in extrapolation tasks.** We validate a *key feature* of our method: the visual graph encoder's ability to construct reward functions for extrapolations of the demonstrated tasks and train agents to manipulate an unseen number of objects. We begin by testing the results of policy learning from videos showcasing cross-embodiment demonstrations of pushing 2 debris. We then increase the

---

[1]For experiment results of X-MAGICAL in domain-shift environment, please see Appendix A.4

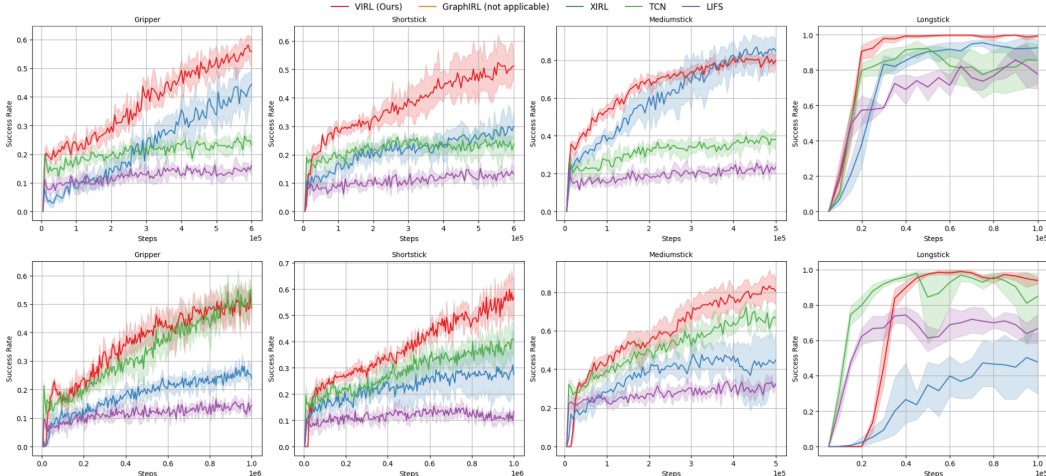

Figure 3: **Cross-embodiment extrapolation tasks.** Learning to push 3 debris given videos of pushing 2 debris (top), and videos of pushing 1 debris (bottom). **Note**: GraphIRL is *not applicable* when agents are learning to manipulate a different number of objects than in the video demonstrations.

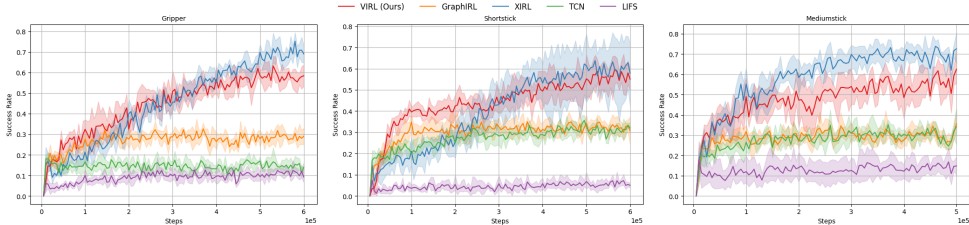

Figure 4: **Cross-embodiment visual-feature-dependent task.** Our approach can be applied for visual-feature-dependent tasks. For example, pushing 2 debris with the identical shape out of 3. **Note**: *long-stick* is *not included* for this task because its excessive length hinders the task execution.

difficulty by providing demonstrations of pushing only 1 debris, as shown in Figure 1a (top). In Figure 3 (top), we illustrate that in the first scenario, our approach outperforms baselines for 3 out of 4 embodiments and is on par with XIRL for medium-stick. In the more challenging scenario, where demonstrators push 1 debris, policy learning becomes significantly harder, as it is not demonstrated that returning to fetch additional debris leads to greater rewards in the long term. Figure 3 (bottom) shows that our approach generalizes to manipulate the unseen number of objects with higher success rates than others. Surprisingly, TCN outperforms XIRL in the most challenging setting and delivers performance comparable to our approach for the gripper and long-stick configurations.

**Results in visual-feature-dependent task.** We evaluate the second *key feature* of our method: the adaptability to visual-feature-dependent tasks by encoding visual features of objects into nodes. The task for imitators to learn is identical to what is demonstrated, as shown in Figure 1b. Hence, it can be anticipated that the performance of XIRL should be comparable to our approach. Figure 4 reveals that our approach is capable of policy learning for geometry-aware tasks, achieving similar performance to XIRL for two embodiments, yet slightly lower for medium-stick. On the other hand, GraphIRL shows inferior performance in this task because of the lack of visual information.

### 4.3 Experiment results on Robot Visual Pusher

**Learning robotic manipulation from human videos.** We build reward functions on encoders pretrained on human videos and use them for policy learning of demonstrated task. Task I in Figure 5 shows the result. Although XIRL can improve the sample efficiency when combined with sparse

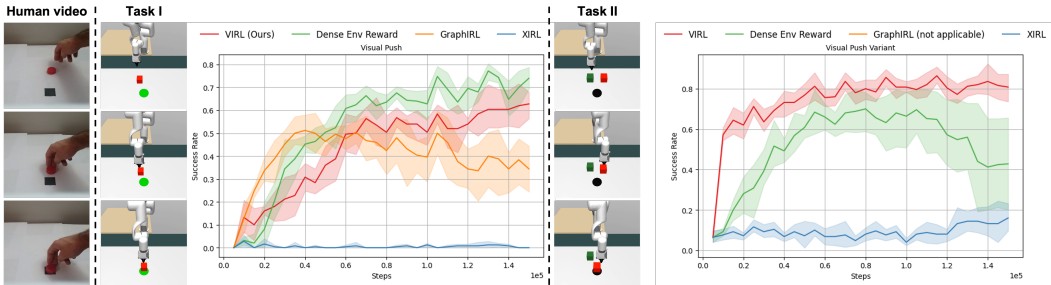

Figure 5: **Robot Visual Pusher.** Task I: robot learns the demonstrated task. Task II: robot learns variant task to push a specific object among 2 objects to goal. Thresholds for success, measured by distance between positions of object and goal, are 5cm for Task I and 10cm [24] for Task II.

environment reward [23], we notice that its learned rewards alone are insufficient for policy learning. In contrast, our approach provides a learned reward function closest to dense environment rewards.

**Learning visual-feature-dependent task variant.** We use the *same encoders* in the first task to build reward functions for the task variant: pushing a specific object among two objects to the goal. It merits attention that GraphIRL is not applicable for this task since it encodes object counts in demonstrations into node features and cannot generalize to scenarios with unseen numbers of objects. The result of learning Task II in Figure 5 shows that our approach VIRL surpasses the performance of using handcrafted dense environment rewards, while it is hard for agents to learn functional policies given reward signals from the baseline XIRL.

Additionally, we ablate the detector's accuracy to study the impact of missing detection in both robot tasks. For experiment details and results, please see Appendix A.5.

**Sim2Real transfer.** We evaluate policies trained in simulation, without domain randomization, on a real-world xArm 6 robot in the Robot Visual Pusher - Task II. Each policy runs for 20 trials: the green object is placed on the left and the red on the right for the first 10 trials, then their positions are switched. Results are in Table 1. The performance gap between simulation and real robot likely stems from unaligned physics properties of objects and lack of domain randomization. For details of robot setup and episode visualization, please see Appendix A.6.

| Method | Success rate |
|---|---|
| VIRL (ours) | **0.55** |
| XIRL | 0.05 |
| Env. reward | 0.2 |
| GraphIRL | - |

Table 1: Real robot test results.

## 5 Discussion

**Conclusion.** We present VIRL, an IRL method utilizing a visual graph encoder to construct reward functions for demonstrated tasks in videos and their extrapolation variants. VIRL integrates object-level spatial and visual features unifiedly into nodes for graph construction from frames. Pretraining the visual graph encoder is self-supervised using TCC loss and reconstruction loss. The design of the visual graph encoder enables effective reward function generation for the demonstrated task and extrapolations involving unseen object numbers, which is challenging in robotic tasks [26]. Additionally, our approach is flexibly adaptable for visual-feature-dependent tasks.

**Limitations.** Our approach depends on reliable object detectors. While occasional detection failures have minimal impact, missing detection in keyframes can be critical. Additionally, the tested tasks don't fully capture the complexity of kinematics and scene variations in benchmarks like RoboSuite [76], MetaWorld [77], and BridgeDataV2 [78]. Third, it remains unclear how VIRL would generalize to deformable object manipulation [79, 80]. Lastly, focusing only on object-centric information may limit performance on tasks requiring broader scene context. [2]

---

[2]For a more comprehensive discussion of limitations, please see Appendix A.9

**Acknowledgments**

We would like to thank the reviewers for the insightful comments and thank Omar Swei, Renjie Liao, Tengda Han, Qi Zhu, Runsheng Xu, and Binghao Huang for the fruitful discussions.

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

# A Appendix

## A.1 Property Comparison between VIRL and Baselines

| Method | Extrapolation task | Visual-feature-dependent task | Task in shifted domain | Cross-embodiment |
|---|---|---|---|---|
| XIRL [23] | × | ✓ | × | ✓ |
| GraphIRL [24] | × | × | ✓ | ✓ |
| VIRL (ours) | ✓ | ✓ | ✓ | ✓ |

Table 2: Property comparison in different task settings between VIRL and the most related works.

## A.2 Extended Related Work

**Imitation learning with actions.** Imitation learning from expert demonstrations with state and action pairs has demonstrated successes in various approaches [27], from behavioral cloning [28, 29, 30] to inverse reinforcement learning [17, 31, 32]. However, its reliance on ground-truth actions renders data acquisition at scale both costly and challenging, thereby limiting its wider applicability. To leverage unlabeled video demonstrations that are easier to obtain, methods have been developed to infer actions in videos [33, 34, 35, 36, 37, 38, 39]. For example, Pathak et al. [34] utilize a forward model so that, given the current state and the policy's predicted action, it outputs the next state. With the state sequences in state-only demonstrations, both the model and the policy can be trained using the difference between the forward model's output and the ground truth of the next state. A similar forward dynamics model is also used in Edwards et al. [36]. In another example, VPT [39] trains an inverse dynamics model (IDM) on a small labeled dataset and then uses the trained IDM to augment a large amount of unlabeled videos with actions, which are used for the subsequent policy learning via behavioral cloning. These methods have successfully trained agents using unlabeled videos, but are subject to well-defined morphologies and dynamics of demonstrators and imitators, making it hard to leverage demonstrations with diverse embodiments.

**Domain adaption for imitation learning.** To overcome the domain gap between varied agent embodiments, several works [40, 41, 42, 43, 44, 18] attempt to translate the context in demonstrations to the imitators' context in terms of agents and viewpoints, using methods such as generative adversarial networks [45, 46] and context translation. Furthermore, another way to address the domain gap between agents is by inpainting them from video demonstrations and online visual states [47].

**Inverse reinforcement learning from videos.** On the other hand, a group of works [21, 22, 48, 49, 50, 44, 51, 52, 53, 24] has focused on learning reward functions from videos by, instead of using any additional models for action prediction or domain translation, extracting latent features that indicate intermediate steps or measure task progress, which are naturally and implicitly present in demonstrations. These learned reward functions are then used for policy training in the regular RL regime. For example, Aytar et al. [48] pretrain a visual encoder on classifying temporal distances between frames in a single video, retrieve a sequence of embedding checkpoints in the video, and then generate reward signals by comparing observations and embedding checkpoints. For learning robotic tasks from human videos, Sermanet et al. [22] train viewpoint-invariant representations on pairs of simultaneous videos with an objective of attracting positive frame pairs while repulsing negative frame pairs in the latent space. They demonstrate that the agent-invariant representations can directly provide reward signals for robot arms. To handle video demonstrations with varied lengths and paces, XIRL [23] adopts temporal cycle-consistency loss [25] to learn visual embeddings that measure task progression and then uses distances between current state embeddings and the goal embedding as reward signal for policy learning via RL. They demonstrate the pretrained encoder benefits from the diversity of demonstrators' embodiments and generalizes to agents with unseen embodiments. Building on XIRL, our work seeks to train an encoder that generalizes to constructing reward functions for task extrapolations by leveraging graph abstractions, while maintaining the flexibility for tasks requiring attention to low-level visual features by encoding object features.

**Object-centric scene graphs.** Abstracting visual scenes into object-centric scene graphs is beneficial for agents to understand environments and make decisions in various tasks [54], such as dynamics modeling [55, 56], navigation [57, 58, 59], and robotic manipulation [60, 61, 62, 43, 63, 24, 64]. As a follow-up work to XIRL, GraphIRL [24] enhances robustness to diverse object appearances by abstracting frames into graphs, where each node represents an object and contains features including its bounding box coordinate and distances to other objects. However, limitations arise when tasks require attention to object geometries or when dealing with different numbers of objects. Motivated by their work and aiming to address the limitations, we encode object visual features and locations into nodes in a unified way and build graphs such that the encoder can generalize to scenes with arbitrary numbers of objects. Our work can be considered a generalized version of GraphIRL, suitable for both visual-feature-dependent and -independent tasks, as well as unseen task extrapolations.

## A.3 Implementation Details

The visual encoder borrows from a pretrained ResNet-18 [81]. We take 512-dimensional embeddings as node features to construct complete graphs and use graph transformer [71] to extract 512-dimensional graph embeddings. We use the graph embeddings for reconstruction from decoder and temporal alignment. We adopt ADAM [82] optimizer and Soft Actor-Critic (SAC) [83] RL algorithm. Hyperparameters are from Zakka et al. [23] with minimum fine-tuning on the number of frames per sequence in pretraining due to memory limitation. Learning rates are $10^{-5}$ for pretraining and $10^{-4}$ for RL. Our approach does not rely on any data augmentation. All results represent the mean performance over 5 random seeds.

## A.4 Additional Experiment Results of X-MAGICAL

**Results in domain-shifted environment.** We assess VIRL's performance in shifted domains, specifically with respect to unseen goal region colors and debris shapes, as depicted in Figure 1c. In contrast to GraphIRL, which enhances robustness to unseen environments by disregarding visual features and focusing solely on graph abstractions of coordinates of bounding boxes, we encode both spatial and visual features into embeddings. Our aim is to validate that graph embeddings remain effective for measuring progress despite the incorporation of unseen visual features. Figure 6 shows that our approach significantly outperforms baselines in the short-stick scenario. For gripper, our approach and GraphIRL achieve similar performance, outperforming the rest, which is expected considering that GraphIRL's abstracted spatial information of objects is sufficient for learning this task and is robust to shifted domains. For medium-stick and long-stick, results show that our approach is on par with the best baselines.

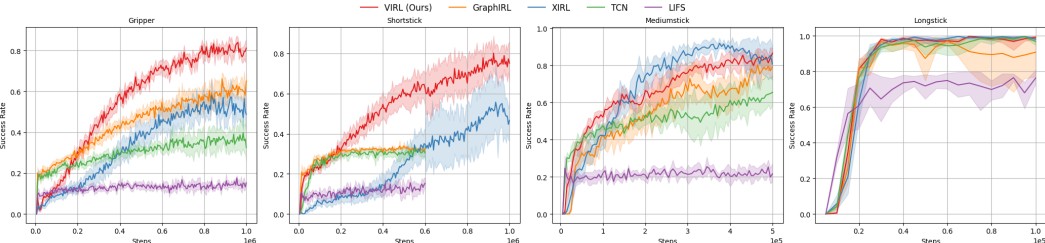

Figure 6: **Cross-embodiment cross-environment.** Encoder is pretrained on demonstrations as shown in Figure 1c (top). Policy learning in diverse environments with randomized goal region color and debris shape as shown in Figure 1c (bottom). Each agent (gripper, short-stick, medium-stick, long-stick) learns from videos of agents with the other 3 embodiments. For example, gripper learns from videos by short-stick, medium-stick, and long-stick.

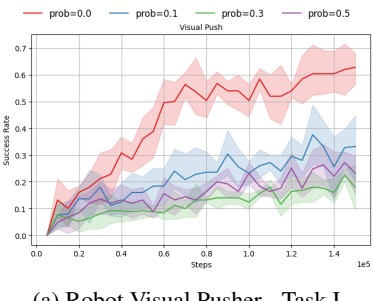
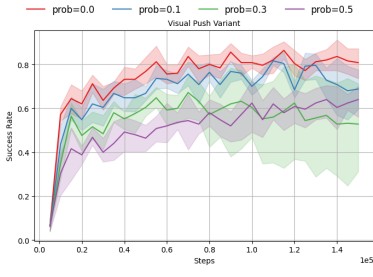

(a) Robot Visual Pusher - Task I          (b) Robot Visual Pusher - Task II

Figure 7: Impact of object detection failures. Each object is set to be not detected with a probability. We test with probabilities being 0.1, 0.3, and 0.5 for both Task I and Task II.

## A.5 Ablation Study

**Impact of missed object detection.** Our approach requires a reliable object detector in both pre-training and policy learning. The detection performance on collected demonstrations can be exhaustively evaluated, while that on scenes during policy learning is prone to uncertainty. Therefore, we study the effect of object detection failures during policy learning. We randomly mask an object's visual and positional features by setting its crop to be zeroed out with a probability. The objects subject to mask include the robot end effector, the red and green cubes, and the goal. We test with probabilities being 0.1, 0.3, and 0.5 for Robot Visual Pusher Task I and Task II, and the results are shown in Figure 7. The effect brought by failing to detect all objects in scene at a step is that the corresponding reward signal may not accurately reflect the actual task progress. Interestingly, performance drop by detection failure varies between tasks.

## A.6 Real-World Robot Setup

In the real-world robot experiment, we apply the trained policies of the variant task of Robot Visual Pusher, i.e., Task II in Figure 5, on a Ufactory xArm 6 robot arm. The observation is captured by a fixed Intel RealSense D435 camera. The robot setup is shown in Figure 8. We utilize the codebase of Diffusion Policy [84, 85] for the policy deployment on the real robot. We also visualize a successful trial and a failed trial as shown in Figure 9a and Figure 9b, respectively.

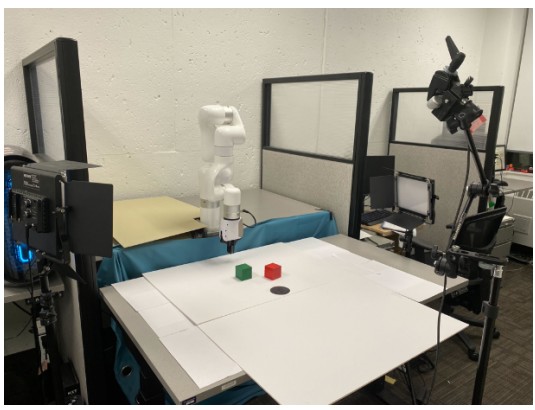

Figure 8: Real-world robot experiment steup.

## A.7 Details of Object Detector

**Detection for X-MAGICAL.** We use YOLO-v8 [86] as the object detector for X-MAGICAL experiments. For encoder pretraining, we train the detector on a small set of images from demonstration

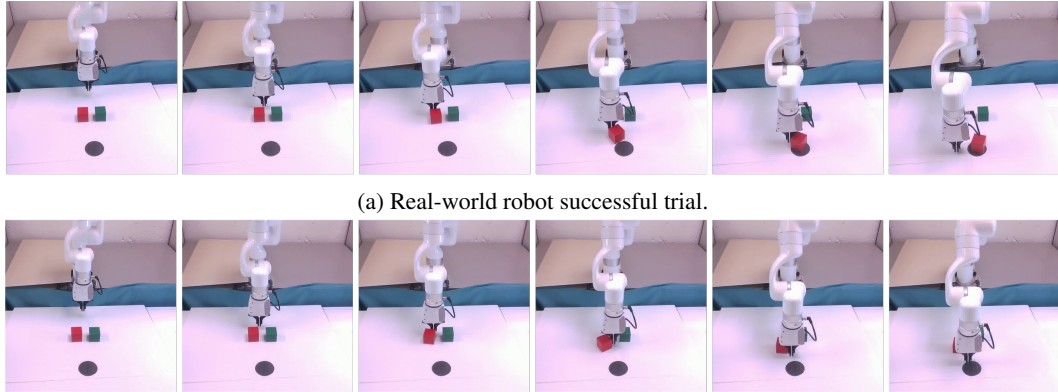

(a) Real-world robot successful trial.

(b) Real-world robot failed trial.

Figure 9: Real-world robot experiment visualization.

data by XIRL [23], and evaluate it to ensure that detection is successful for all demonstration videos. For policy learning, we adopt the trained detector from the pretraining step. Given an image, we apply the detector to detect objects and build the corresponding graph of the image. However, the detection is not always successful during policy learning. We use the image from the previous step when detection failure happens.

**Detection for Robot Visual Pusher.** We also use YOLO-v8 [86] as the object detector for Robot Visual Pusher experiments, but only in pretraining. Similarly, for encoder pretraining, we train the detector on a small set of images from demonstration in [44], and evaluate it for successful detection of the object and the robot end effector over all demonstration videos. For scenes where the goal is occluded by the object and cannot be detected, we set the goal bounding box to be the summation of the object bounding box and Gaussian noise. For policy learning, we do not use any object detector. Instead, we follow GraphIRL [24] and compute objects bounding boxes in 2D image.

## A.8 Technical Comparison between VIRL and GraphIRL

VIRL differs from and improves upon GraphIRL [24] mainly in three aspects: (i) the construction of graphs from images, (ii) the choice of graph encoder, and (iii) the loss function for pretraining.

**Graph construction.** GraphIRL concatenates an object's bounding box coordinate with its distances to other detected objects as the node feature $\{x_1, y_1, x_2, y_2, d_1, d_2, ..., d_{N-1}\}$. Although robust to domain shift and visual distraction, their design has two downsides: First, it keeps object spatial information via coordinates while disregarding the visual features; Second, a pretrained graph encoder cannot generalize to policy learning for tasks where the number of objects is different from that in demonstrations because of the change of node feature length.

To resolve the first issue, VIRL encodes object-wise visual features into graph nodes. Specifically, VIRL creates an image for each detected object by zeroing out pixels outside of the object bounding box, keeping images shape identical to the original scene image. We then use a convolutional neural network to extract the object's visual feature $\mathbf{f}_{o_i} = \text{ResNet}(I_o)$. In this way, we make our approach applicable for tasks dependent on visual features. To resolve the second issue, we simply make the distance between two nodes the feature of the edge connecting them.

**Graph encoder.** GraphIRL proposes a Spatial Interaction Encoder for their graphs following [55] and [87] to extract abstract information. In contrast, we adopt Graph Transformer [71] as the graph encoder. We argue that any graph neural network that is capable of extracting decent global features from graphs with edges should be applicable in our approach.

**Loss function for pretraining.** GraphIRL follows XIRL [23] and uses Temporal Cycle-Consistency (TCC) loss as the loss function for pretraining their graph encoder. In contrast, we follow [73] and

add a reconstruction loss apart from TCC loss to ensure that sufficient object-centric low-level visual features are kept in the graph embedding for learning policies of visual-feature-dependent tasks.

## A.9 Extended Limitations and Opportunities

First, our approach relies on reliable object detectors. Although occasional nonconsecutive detection failure may result in limited influence, detection failure in keyframes can be influential. The opportunity brought by this limitation is that our approach may be applicable for more complex tasks by leveraging more powerful open-vocabulary detectors, such as OWL-ViT [88] and YOLO-World [89]. By incorporating object tracking methods, the occlusion cases may be also improved. Second, while our approach is tested on X-MAGICAL, Robot Visual Pusher, and real-robot, which provided valuable initial insights, we acknowledge that it does not fully capture the complexity of kinematics and scene variations present in more challenging benchmarks, such as tasks in RoboSuite [76], Meta-World [77], and BridgeDataV2 [78]. Third, our approach is proposed for rigid object manipulation in the first place. Thus, it is unclear how it would generalize to tasks of deformable object manipulation [79, 80]. Fourth, since our approach focuses on object-centric information while disregarding non-object-centric information, it might struggle with certain tasks that require an understanding of the scene context.

