# OpenReview forum: "VIRL: Self-Supervised Visual Graph Inverse Reinforcement Learning"
_robot-learning.org/CoRL/2024/Conference — CoRL 2024_

### Official Review · Reviewer_SzPu · 2024-07-11

**Originality:** 3
**Technical Quality:** 3
**Clarity Of Presentation:** 4
**Potential Impact:** 3
**Recommendation:** 3
**Confidence:** 4

**Review:**

Strengths
+ The approach is reasonable and seems to work.  The generalization to unseen numbers of objects is nice.

+ While there are no hardware experiments, the paper transfers from real human demonstrations to simulated robot environments, which indicates that it should be sufficiently visually robust to be applied to some real world robotic tasks.


Weaknesses
- There are no ablations of the proposed method.  What design choices are important?  It would be good to include some ablations of things like the loss functions, positional embeddings, and the process for extracting object-level crop features.

- The experiments are all in fairly simple tasks that involve relocating objects.  How well does this approach scale to more complicated tasks, especially tasks that involve changing the state or orientation of an object, tasks that involve occlusions, and tasks that involve dynamic motion?

- The approach discards all of the non-object-centric information in the scene, which may be important for certain tasks.

**Quality Of The Limitations Section:**

3

**Questions For Rebuttal:**

How well does the method handle objects with overlapping bounding boxes?

What off-the-shelf object detector is used by this method?

**Robotics Focus:**

3

**Summary Of Paper:**

The paper presents an approach to inverse reinforcement learning from video demonstrations.  The approach first encodes the video demonstrations into an object-centric graph representations.  The graph embedding is pretrained with a cycle-consistiency loss and an object-centric reconstruction loss.  The resulting embedding can be used as a reward function by calculating the distance between the current embedding and the goal embedding.

**Summary Of Recommendation:**

Weak Accept:  This paper is interesting and proposes a novel method, but it is unclear how well it will work on more complicated tasks.

---

### Official Review · Reviewer_M2rM · 2024-07-18

**Originality:** 2
**Technical Quality:** 2
**Clarity Of Presentation:** 2
**Potential Impact:** 3
**Recommendation:** 3
**Confidence:** 4

**Review:**

# Strengths

1. The integration of visual features and graph abstractions is a significant strength.

2. The self-supervised approach for pretraining the visual graph encoder is good. It reduces the need for extensive manual labeling and allows the model to learn from unlabeled video data effectively.

3. The paper demonstrates that VIRL can handle tasks necessitating both granular visual attention and broader global feature consideration. This is evident in its performance on the X-MAGICAL and Robot Visual Pusher benchmarks.


# Weaknesses

1. VIRL still has the reliance on object detectors. This limits its wide applicability in more complex tasks.

2. While the paper demonstrates strong performance in simulation environments, there is a lack of testing on physical robots. The transfer of simulation results to real-world scenarios is not guaranteed and would be crucial.

3. This paper does not clearly state the technical difference between it and GraphIRL but only claims the difference in motivation.

**Quality Of The Limitations Section:**

3

**Questions For Rebuttal:**

1. The curves in Figure 6 are not converged. It is good to see a better sample efficiency, yet it would be also interesting to see the converged results. Could authors run experiments to converge and show the results?
2. Showing experiment results on real robots would give better insight into how VIRL helps real-world robot learning.
3. Detailed difference between GraphIRL. What is the actual technical contribution that differs VIRL from GraphIRL?
4. No ablation studies on the key factors that make GraphIRL perform better than previous methods.
5. Figure 2 could be largely improved for better visualizations.

**Robotics Focus:**

3

**Summary Of Paper:**

This paper proposes a generalized version of GraphIRL, named VIRL,  which pre-trains a graph encoder on videos and then applies that encoder to extract reward functions for downstream tasks.  VIRL shows better results over previous methods such as XIRL on in-domain tasks and the extrapolation ability.

**Summary Of Recommendation:**

I would recommend the rejection for the current version. I think the idea of generalizing reward function learning is good, but the current paper seems to not propose too much new insights/techniques compared to previous methods. Also, the experiments such as ablation studies and real robot experiments are lacking.

---

### Official Review · Reviewer_4TrT · 2024-07-20

**Originality:** 3
**Technical Quality:** 3
**Clarity Of Presentation:** 4
**Potential Impact:** 3
**Recommendation:** 3
**Confidence:** 4

**Review:**

## Strengths

- Although the method involves quite a few steps, it’s well-explained and easy to understand. The figure is well-made and helpful to illustrate the approach
- The assumptions for training the approach seem not too strong such that it could be tried on more realistic datasets
- The results are cool — the method can generalize the learned distance function to new embodiments and scenes

 ## Weaknesses

A) Not robust to moving distractors: the method does not distinguish between task-relevant objects and task-irrelevant objects but instead the learned representation that is used to compute distance rewards captures all objects in the scene. This would fail if eg in a driving scene other cars are moving in the background but aren’t relevant to the task. The ego vehicle would still learn to optimize it’s behavior such that the background cars match the behavior from the demonstrations

B) Clean evaluation environments: relatedly, all tested environments are very clean with only task-relevant objects, no occlusions, no mis-detections, no real robots. Learning and computing distances on a more realistic dataset could be informative.

C) Needs object detections: the title says “self-supervised” but in reality a lot of the generalization benefits come from an object-centric representation which is obtained using (supervised) object detectors. Thus, the method will always be reliant on good pre-trained object detectors, and may not work well if they fail (and they do fail eg if there are many smaller objects in a scene that the agent is trying to manipulate.

D) The current approach is very much designed with the idea of rigid object manipulation in mind. It is unclear how it would generalize to eg manipulating cloth since individual parts of the cloth would not be detected as separate objects.

**Quality Of The Limitations Section:**

3

**Questions For Rebuttal:**

- it would be great to add an experiment that evaluates the method on a less clean, real world dataset like BridgeDataV2

**Robotics Focus:**

3

**Summary Of Paper:**

The paper proposes an approach for learning reward functions with little supervision using a graph-based encoder network and temporal consistency and reconstruction objectives. The final reward function can generalize to computing meaningful distance rewards in scenes with more objects or even from human videos to robot scenes.

**Summary Of Recommendation:**

Overall, I like the paper and I recommend accepting it. I mentioned multiple downsides but I don’t think they outweigh the interestingness of the approach. If possible, I would encourage the authors to evaluate their approach on some messy real robot data like BridgeDataV2 with a lot more distractors and visual diversity.

---

### Official Review · Reviewer_avVr · 2024-07-24
**Paper Review**

**Originality:** 3
**Technical Quality:** 3
**Clarity Of Presentation:** 4
**Potential Impact:** 2
**Recommendation:** 3
**Confidence:** 4

**Review:**

Strengths:
1. This paper is well-written and very clear to follow.

2. The method is simple and builds on several prior works; it is likely to work on environments not demonstrated in the paper.

Weaknesses:
1. The proposed method seems to be specific for tasks that require generalization over object counts. VIRL does not seem to outperform XIRL on visual-feature-dependent task as in Figure 4. Furthermore, its main advantage over GraphIRL appears to be just the extrapolation capability to unseen number of objects from the graph transformer encoder design.

2. No real-world results are demonstrated; in contrast, the main baselines, GraphIRL and TCN, in the paper have demonstrated hardware results.

3. Relatedly, the tasks are quite simple kinematically and assume full-observability. Experiments on more manipulation-centric benchmark, such as RoboSuite or MetaWorld, could make more convincing case of the utility of the proposed method.

4. The paper lacks an ablation section; Specifically, given that the method relies on a object detector, it would be good to know whether the method still works when the object detector is not always accurate.

Overall, I think this is a paper with moderate contributions that could be useful to the broader community on inverse visual RL for robotic control; however, the method could benefit from more convincing real-world results.

**Quality Of The Limitations Section:**

3

**Questions For Rebuttal:**

My concerns are stated above. Here are some additional minor comments:
1. The texts in Figure 2 and 3 are hard to read; consider re-sizing.

**Robotics Focus:**

3

**Summary Of Paper:**

This paper introduces VIRL, a novel graph object-centric visual representation that can be used for inverse RL. The main advantage of VIRL is its ability to generalize to unseen number of objects. Simulation results are shown on the X-Magical benchmark as well as a visual pushing task.

**Summary Of Recommendation:**

Weak Accept because this paper is technically correct but moderate in impact.

---

### Author Rebuttal · Authors · 2024-08-13

We sincerely appreciate the reviewers and the meta-reviewer for their thorough and constructive responses, which inspire us to think about more interesting scenarios and conduct additional experiments. Based on your feedback, we have made the following changes and updates to the revised manuscript:

* Added real-world robot experiment for the variant task of Robot Visual Pusher. The success rates are reported in Table 1 in Section 4.3.

* Added real-world robot experiment setup introduction and visualization in Appendix A.4 .

* Added ablation study on the impact of object detection failures in Section 4.4.

* Added Table 2 in Appendix A.1 to display the property advantages of our approach over GraphIRL and XIRL in a clearer way.

* Improved Figure 2 for better visualizations and readability.

* Added Appendix A.6 for discussing technical differences between our approach and GraphIRL.

* Updated Figure 5 - Task II in Section 4.3: Experiment results on Robot Visual Pusher. We reran policy learning by VIRL for Robot Visual Pusher - Task II after fixing a bug in the code.

* Updated the discussion of limitations in Section 5 to make it more comprehensive.

* Updated Figure 7 (which used to be Figure 6) to see X-MAGICAL domain-shift experiment results after convergence. We extend the training steps for gripper and short-stick from $6\times10^5$ steps to $1\times10^6$ steps.

* Added the details of object detection used in this paper in Appendix A.5.

---

### Decision · Program_Chairs · 2024-09-04

**Decision:**

Accept

**Comment:**

The submission proposes an extension of graph IRL to pretrain on visual data with detailed experiments and analysis.

The submission is well-written and presents a clear and simple, general method. It demonstrates promising results in generalizing learned distance functions to new embodiments and scenes. The integration of visual features and graph abstractions is a significant strength, and the self-supervised approach for pretraining is a good idea.

The proposed method seems to struggle with moving distractors and is strongly affected by noise in perception and detection. This limits its application wrt complex scenarios with occlusions or mis-detections. The lack of real-world robot experiments and exploration of more manipulation-centric benchmarks limit its convincingness. Additionally, the approach discards non-object-centric information, which might be important for certain tasks. The paper could benefit from including ablations and addressing the scalability of the method to more complex tasks.

The paper has improved on the above description during the rebuttal via detailed communication between reviewers and authors. Based on the above the paper is recommended for acceptance. Please follow up on the remaining open points for a potential camera ready version.